# The Germination and Growth of Two Strains of *Bacillus cereus* in Selected Hot Dishes After Cooking

**DOI:** 10.3390/foods14020194

**Published:** 2025-01-09

**Authors:** Josef Kameník, Marta Dušková, Alena Zouharová, Michaela Čutová, Kateřina Dorotíková, Michaela Králová, Blanka Macharáčková, Radka Hulánková

**Affiliations:** Department of Animal Origin Food and Gastronomic Sciences, Faculty of Veterinary Hygiene and Ecology, University of Veterinary Sciences Brno, Palackého tř. 1946/1, 612 42 Brno, Czech Republic; kamenikj@vfu.cz (J.K.); skockovaa@vfu.cz (A.Z.); cutovam@vfu.cz (M.Č.); katerina.dorotikova@delirest.cz (K.D.); kralovam@vfu.cz (M.K.); macharackovab@vfu.cz (B.M.); hulankovar@vfu.cz (R.H.)

**Keywords:** artificial contamination, *Bacillus cereus* spores, hot dishes, pH value

## Abstract

The aim of this study was to assess the germination and growth of two strains of *Bacillus cereus* following the artificial inoculation of six selected hot dishes with spores which were then stored at temperatures of 40, 50, and 60 °C for 0.5, 1.0, 2.0, 2.5, 3.0, and 4.0 h. The water activity of the prepared meals varied between 0.967 and 0.973 and the salt content between 0.74 and 1.40%. The pH value of four dishes exceeded 6.0, but for two (tomato sauce and ratatouille) it was 4.6. The tested strain DSM 4312 showed good growth abilities and attained a population exceeding 6.0 log CFU/g within 4 h at 40 °C in foods with pH values > 6.0. The study demonstrated that a drop in food temperatures to 40 °C is risky, while no growth of *B. cereus* was detected within 4 h at 50 and 60 °C. The growth rate of *B. cereus* is conditioned not merely by environmental conditions (temperature, pH values, food composition), but also by the bacterial strain.

## 1. Introduction

*Bacillus cereus* has become a problem of increasing importance in the food industry in terms of hygiene and technology [1,2]. This sporogenous bacterium is widespread in the environment. It is found in soil and water, on the surface of plants, and in the rhizosphere [3]. Vegetative cells and, in particular, spores can easily enter the food chain through food crops [2].

There are two types of foodborne illness caused by strains of *B. cereus* that differ in relation to both their symptoms and their origin. They are known as emetic syndrome and diarrheal syndrome [4]. Although *B. cereus* is recognized as one of the leading causes of foodborne diseases, the true number of cases caused by *B. cereus* is unknown due to the lack of systematically collected data. The official statistics for all foodborne outbreaks caused by *B. cereus* underestimate the true extent of the problem and may represent only around ten percent of the actual number of cases [5]. This is due to the relatively short duration of both forms of the disease and—with exceptions—the rapid disappearance of their symptoms [6]. In member states of the European Union, 306 foodborne outbreaks caused by *B. cereus* toxins were recorded in 2022 [7]. This was an increase of 251.7% as compared to 2021, and *B. cereus* caused five extremely large outbreaks in France and Spain in which a hundred or more cases/outbreak were demonstrated [7]. Foods in the category mixed food, among which ready meals rank in first place, were identified as the vehicle.

To prevent the growth of microorganisms in food, the generally accepted “40–140” rule recommends that food should be kept at temperatures either below 40 °F (4.4 °C) or above 140 °F (60 °C) for reasons of food safety [8]. Ready meals are among the foods in which the risk of contamination or the survival of foodborne pathogens is high. Ready meals are cooked dishes intended for direct consumption that are divided into two large groups according to the Czech legislation [9]. Hot dishes are distributed in such a way as to reach the consumer as quickly as possible at a temperature of at least 60 °C [10], and this also meets Food and Drug Administration (FDA) requirements for maintaining hot foods at temperatures of 140 °F or higher [11]. Cold dishes are distributed on the market at temperatures of no more than 8 °C [12]. Because *B. cereus* spores can survive the cooking process (a D_90°C_ value of between 5 and 102 min is reported by Webb et al. [13]), a decisive role is played in hot dishes by their temperature and the period of time from preparation to consumption. *B. cereus* ceases to grow at temperatures of 48 °C or 55 °C [14] and at pH values < 4.3 depending on the phylogenetic group [13].

The delivery of ready meals by courier services has become widespread in large cities in Europe in recent years [15,16,17]. While a minimum temperature of 60 °C can be achieved when serving dishes in catering service establishments, this is practically impossible when distributing food over longer distances associated with a longer time interval. There may, therefore, be a risk of the growth of undesirable bacteria, including *B. cereus*.

The aim of this study was to assess the germination and growth of two strains of *B. cereus* following the artificial inoculation with spores of six selected hot dishes stored at temperatures of 40, 50, and 60 °C.

## 2. Materials and Methods

### 2.1. Test Batches of Hot Ready Meals

Mashed potatoes, mushroom sauce, tomato sauce, and cooked rice were prepared according to the recipes by Runštuk et al. [18], bucatini carbonara according to Bardi [19], and ratatouille according to Roux [20]. The composition of the dishes is shown in Table 1. All the listed dishes belonged to the category of hot ready-to eat foods and a cooking temperature of at least 90 °C was attained during their preparation. A Testo 104-IR thermometer (Testo, Prague, Czech Republic) was used for testing. These test batches were prepared for the purpose of setting-up food preparation technology and for the analysis of natural contamination with *B. cereus* (Figure 1, Appendix A).

### 2.2. Hot Ready Meals for the Inoculation with Bacillus cereus Spores

The dishes (Figure 1) were prepared over the period from 06/2023 to 04/2024 and inoculated with a suspension of *B. cereus* spores using the CCM 869 (CCM, Brno, Czech Republic) or the DSM 4312 (DSMZ, Braunschweig, Germany) strains. *Bacillus cereus* CCM 869 is a diarrheal strain [21] and *B. cereus* DSM 4312 is an emetic strain [22]. The prepared dishes were contaminated with a suspension of *B. cereus* spores to a level of 2–4 log CFU/g of food. The spores in the contaminated food were activated by heating in a water bath (70 °C/5 min). Individual portions (150 g) were placed in polypropylene trays and packed with no change of atmosphere in a T-190 sealing machine (MetalPack, supplier Maso-profit, s.r.o., Prague, Czech Republic) with a 185 mm upper PET/PP foil (thickness 52 μm; supplier Maso-profit s.r.o.) at a sealing temperature of 180 °C. This was followed by tempering in incubators to temperatures of 40, 50, and 60 °C and storage at these temperatures. Individual samples were taken over the course of 4 h and the growth activity of the *B. cereus* strains used was monitored. Samples were taken for microbiological testing immediately after packaging and then after 0.5, 1.0, 2.0, 2.5, 3.0, and 4.0 h. A non-inoculated control of the given dish was also tested in parallel. The experiments were performed in duplicate with the preparation of two time-separated batches.

To analyze the effect of pH values on the growth of *B. cereus*, the pH values of mashed potatoes were adjusted to 4.3 using glacial acetic acid, and tomato sauce to a value >6.0 using 10 M NaOH. The effect of adjusted pH values was tested only in these two selected dishes, which were stored at 40 °C and samples were taken after 2.0, 3.0, and 4.0 h. (Figure 1).

### 2.3. The Preparation of Spore Suspensions

Selected strains of *B. cereus* were cultured in buffered peptone water (BPW; OXOID, Basingstoke, UK) to obtain a culture with exponential growth with a density of approximately 7 log CFU/mL (1 mL of inoculum with a density of 1 McFarland Unit was added to 9 mL of BPW and cultured at 30 °C/24 h/aerobically). Five-hundred microliters of this bacterial suspension were applied to a Petri dish with TGYE agar (tryptone, glucose, yeast extract agar; OXOID) and spread over the entire surface (any excess liquid was pipetted off). The plates were incubated for 10 days at 30 °C/aerobically.

The colonies of sporulated *B. cereus* cells grown were transferred quantitatively with a bacteriological loop into test tubes with 5 mL of physiological solution and centrifuged (3010× *g*/10 min). After centrifugation, the supernatant was decanted into a waste receptacle and 5 mL of physiological solution, in which the sediment was mixed into a homogeneous suspension, and was again added to the test tubes. The test tubes were again centrifuged and the supernatant decanted. This washing was repeated once more (3× in total). After washing, 5 mL of physiological solution was added to the sediment and a homogeneous spore suspension was formed. This suspension was transferred aseptically to an Erlenmayer flask, made up to a final volume of 50 mL with the addition of a physiological solution, and briefly mixed in a vortex. The suspension obtained was treated with ultrasound for 10 min, thereby inactivating any vegetative cells. The spore suspension prepared in this way was divided into a number of aliquots (5 mL each) and stored frozen at a temperature of −80 °C. Every time a new batch of spore suspension was made, the number of spores in this suspension was determined by cultivation to enable the calculation of the correct dilution for the inoculation of the food. Spore formation was examined microscopically (by hot staining with 5% malachite green and carbolfuchsin staining) [23].

### 2.4. Microbiological Analyses

Only *B. cereus* was analyzed in the test batches. The *B. cereus* count, the total viable count, and the count of *Escherichia coli* were determined in the samples of hot ready meals prepared for the inoculation with *Bacillus cereus* spores. Twenty-five grams of sample was weighed into sterile homogenization bags using sterile instruments. The samples were diluted 1:9 with buffered peptone water and homogenized in a Stomacher Star Blender LB 400 (VWR, Radnor, PA, USA) for 90 s. Further decimal dilutions of the samples were prepared as needed from this primary dilution (homogenate).

Mannitol Yolk Polymyxin B agar (MYP agar; OXOID) according to ISO 7932:2004 [24] was used for the determination of *B. cereus* at 30 °C/24 h. Qualitative determination of *B. cereus* was performed by inoculation from the homogenate after 24 h of cultivation at 30° C on MYP agar (30 °C/24 h). Suspected colonies of *B. cereus* were reinoculated onto blood agar and isolates with complete hemolysis were identified by MALDI-TOF MS mass spectrometry and by PCR species-specific identification.

Determination of the total viable count was performed in accordance with EN ISO 4833-1:2013 [25]. The samples were cultured at 30 °C for 72 h under aerobic conditions. Plate count agar (TGYE) with tryptone, glucose, and yeast extract was used. The result was the determination of the number of log CFU of mesophilic aerobic and facultative anaerobic bacteria per gram of sample.

*E. coli* enumeration was performed according to ISO 16649-2:2001 [26] on selective diagnostic agar with tryptone, bile salts, and glucuronide (TBX agar; OXOID, UK) at 44 °C. Qualitative determination of *E. coli* was performed by inoculation of the multiplied homogenate cultured at 44 °C/24 h on TBX agar (44 °C/24 h). Suspected colonies of *E. coli* were identified using MALDI-TOF MS mass spectrometry.

### 2.5. Identification of Bacterial Isolates by MALDI-TOF MS Mass Spectrometry

Following application on a MALDI plate in duplicate, individual colonies of suspected *B. cereus* isolates were overlaid with 1 µL of HCCA matrix (a saturated solution of α-cyano-4-hydroxycinnamic acid in 50% acetonitrile, 47.5% water, and 2.5% trifluoroacetic acid; Merck KGaA, Darmstadt, Germany). Suspected isolates of *B. cereus* were overlaid with 1 µL of 70% formic acid and allowed to dry before the use of the matrix. After drying the matrix, the isolates were subjected to MALDI-TOF MS analysis (UltrafleXtreme instrument, Bruker Daltonik, Bremen, Germany; FlexControl 3.4 software; BioTyper software version 3.0; Bruker Daltonik; BioTyper database entries version 10.0). Only identification outputs with a BioTyper log(score) above 2.0, which means highly reliable identification at the species level, were taken into consideration.

### 2.6. PCR Species-Specific Identification

A polymerase chain reaction (PCR) was used for *B. cereus* sensu stricto (*B. cereus* s. s.) confirmation. Colonies grown on TGYE agar under cultivation conditions of 30 °C/24 h were used for the isolation of DNA by heat. Briefly, one bacterial colony was suspended in 100 µL of sterile physiological solution and heated at 100 °C/10 min, followed by centrifugation at 2082× *g*/1 min. The supernatant was then transferred to a new microcentrifuge tube and an aliquot of 2 μL was used as template DNA in the PCRs. The species identification of *B. cereus* s. s. was performed on the basis of detection of the *gyr*B gene encoding DNA gyrase subunit B with used primers BC1 [5′-ATTGGTGACACCGATCAAACA-3′] and BC2 [5′-TCATACGTATGGATGTTATTC-3′] [27]. The amplified products were separated by electrophoresis on a 2% agarose gel in 0.5× TBE buffer. The gels were stained with Midori Green (Nippon Genetics, Düren, Germany) and visualized using a UV transilluminator (VWR, Radnor, PA, USA).

### 2.7. Physico-Chemical Analyses

The pH value was measured in an aqueous solution of the sample (1:10) using a combined 211 electrode and pH meter (Hanna Instruments, Smithfield, RI, USA) at a temperature of 25 ± 1 °C. The water activity (a_w_) was determined in a well-homogenized sample using a LabMaster aw-meter (Novasina AG, Lachen, Switzerland) at a temperature of 25.0 ± 0.1 °C.

The sodium content was determined by atomic absorption spectrometry. To digest the sample product, 6 mL of concentrated nitric acid (65% *v*/*v*) and 2 mL of hydrogen peroxide (30% *v*/*v*) were added to 0.25 g of the sample and mineralized using an Ethos SEL Microwave Labstation (Milestone, Schio VI, Italy) at 200 °C for 30 min. The sodium content was then measured using air-acetylene flame atomization in a contrAA 700 atomic absorption spectrometer (Analytik Jena, Jena, Germany). All samples were measured in triplicate and the values obtained were processed by Aspect CS software, version 2.1, resulting in one final value for each sample (batch of product). The Na-based salt content (in %) was calculated by applying a conversion coefficient of 2.5 in accordance with Regulation No. 1169/2011 [28].

### 2.8. Statistical Analysis

All data were entered into spreadsheets (Microsoft Office Excel 2019). The obtained experimental data (CFU/g) were log transformed, and the mean values and standard deviations were calculated. The factors dish, temperature, strain, and time were evaluated using a multi-factor ANOVA with interactions followed by the Tukey HSD test for multiple pairwise comparison, using the statistical program Statistica v.7.1. (StatSoft, Prague, Czech Republic). A *p*-value of less than 0.05 was considered to be statistically significant.

## 3. Results and Discussion

### 3.1. Physico-Chemical Parameters of the Prepared Dishes

As shown in Table 2, all the prepared dishes were far from containing the proportion of salt that is limiting for the growth of *B. cereus*. *Bacillus cereus* strains differ in their sensitivity to environmental factors in relation to their classification into phylogenetic groups [13]. Webb et al. [13] reported a proportion of 6% NaCl as representing the growth limit for *B. cereus* s. s., phylogenetic group VI (the limit is 8% NaCl for *B. cereus* s. s. group II and V, or 10% NaCl for *B. cereus* s. s. group III and IV).

The values of water activity (a_w_) that limit the growth of *B. cereus* begin from 0.965 (phylogenetic group I) to 0.960 (phylogenetic group VI). Of the prepared dishes, the lowest value was found for ratatouille (0.967). However, this value allows the growth of all phylogenetic groups of *B. cereus* [13]. As far as pH values are concerned, the limit value for phylogenetic groups I, IV, V, and VI is pH 4.6, while for groups II, III, and VII it is pH 4.3 [13]. From this point of view, the ratatouille and tomato sauce dishes with pH values of 4.6 could prevent the growth of *B. cereus* even under otherwise favorable temperature conditions. A temperature of 40 °C is the limit value for group VI [13], while the limit values for the other groups are temperatures of 43 °C (I, II, V) and 48 °C (III and VI; [13]). Some strains of *B. cereus* are able to grow at temperatures of up to 55 °C [29].

### 3.2. Detection of B. cereus in Test Batches of Dishes

In test batches of hot dishes, *B. cereus* could not be isolated by direct plate counting, because the amount of these bacteria in this samples was below the detection limit (1.7 log CFU/g). With the exception of the cooked rice, *B. cereus* was detected only after enrichment. *B. cereus* could not be detected in samples of cooked rice even after enrichment. All suspected isolates were identified by both MALDI-TOF MS and PCR.

The detection of *B. cereus* in the prepared dishes is not surprising. Despite all efforts, *B. cereus* is still a frequent contaminant of a wide range of foods and their ingredients, including rice, dairy products, spices, dried foods, and vegetables [30,31]. Rahnama et al. [32] performed a meta-analysis of the incidence of *B. cereus* in foods. The overall occurrence of *B. cereus* was 23.7%, while it varied according to the type of food or dish, the detection method, and individual continents. Yu et al. [33] found a 35% prevalence of *B. cereus* in ready-to-eat foods in China.

The absence of *B. cereus* in cooked rice in this study is not unusual. *B. cereus* was detected in 34 out of 100 samples of cooked rice collected in catering service establishments in the study by Navaneethan and Effarizah [34]. Following up from that study [34], the same authors turned their attention to the determination of the effect of heat treatment on the survival and growth of spores of eight different strains of *B. cereus* in rice [35]. Following inoculation and heat treatment (15 min), the samples were stored at temperatures of 4 °C, 25 °C, and 30 °C for periods of 24 h, 48 h, and 7 days. The authors found that the ability of the spores to survive heat treatment is dependent on the particular strain of *B. cereus.* Four of the eight strains used showed no growth or only limited growth during the application of all combinations of temperature and storage period. These were strains that were originally isolated from ready-to-eat cooked rice [34] and had a limited ability to survive heat treatment, which highlights the probability of their origin as contaminants only after cooking.

### 3.3. The Growth of B. cereus Following Artificial Contamination with Collection Strains

Hot dishes must be kept at a temperature of at least 60 °C, which inhibits the growth of *B. cereus* [36]. The requirement for a minimum temperature of 60 °C when distributing hot dishes on the market is defined in the Czech Republic by Decree No. 121/2023 on requirements for ready meals [12]. While it is technically not a problem to ensure this temperature when serving food in catering service establishments, it may not be easy in the case of food delivery by courier services, particularly during the rush hour [37]. In these cases, there may be a risk that the temperature of the delivered food falls beneath the prescribed 60 °C and that, depending on the period of time that has elapsed, the growth of spore-forming bacteria that have survived cooking, including *B. cereus*, may occur. Turner et al. [36] recommend that this period of time should be a maximum of 2 h for potato dishes when the temperature drops below 60 °C.

No total viable counts and *E. coli* were detected in any of the pre-inoculation dishes. Detailed results of microbiological analyses are shown in Appendix A.

#### 3.3.1. Mashed Potatoes

*B. cereus* grows at favorable temperatures in mashed potatoes, with a generation time that may be even less than 1 h [36]. *B. cereus* has been widely isolated from various potato-based products [38].

As can be seen in Table 3, *B. cereus* survived heat treatment and was detected even in the uninoculated batch, in which it reached 1.5 log CFU/g after 2 h at 40 °C and as much as 3.9 log CFU/g after 4 h. Given that the infectious dose of *B. cereus* is 5–7 log CFU (total ingested) for diarrheal syndrome and 5 log CFU/g of food for emetic syndrome [39], a period of just a few hours at a temperature of 40 °C is enough for the bacteria to reach the risk level. The growth of *B. cereus* was not detected in the artificially contaminated batches of mashed potatoes at either 50 °C or 60 °C, and *B. cereus* could not be demonstrated by plate counting in the control batch.

Beside temperature (*p* < 0.001), time (*p* = 0.005) and their interaction (*p* = 0.003) were statistically significant factors, as the increase during incubation was detected only at a temperature of 40 °C. Furthermore, the DSM 4312 strain demonstrated its potential to multiply at this temperature, reaching significantly higher counts at the end of the storage period than the CCM 869 strain (*p* < 0.001).

#### 3.3.2. Cooked Rice

Both strain (*p* < 0.001) and temperature (*p* = 0.013) were significant factors, including their interaction *(p* = 0.007), as only the DSM 4312 strain was able to significantly proliferate and only in samples kept at 40 °C. Juneja et al. [14] did not record growth of a cocktail of emetic and diarrheal *B. cereus* strains in cooked rice at temperatures of 10 or 49 °C, though the bacteria did multiply within the temperature range 13–46 °C. Even in this study, no growth was seen at temperatures of 50 or 60 °C (Table 3). After 4 h at 40 °C, the growth of *B. cereus* was recorded in the control batch without artificial inoculation. Mohammadi et al. [40] also recorded the growth of *B. cereus* in samples of fresh rice noodles without previous inoculation at 32 °C, recording a population of more than 2 log CFU/g after 4 h. In contrast, Juneja et al. [14] did not detect any *B. cereus* bacteria in samples of uninoculated cooked rice.

The differing growth abilities of the *B. cereus* strains used in this study are interesting. While the emetic strain DSM 4312 showed growth at 40 °C after less than 3 h and attained an increase of 3 log CFU within 4 h, the diarrheal strain CCM 869 showed practically no growth and its count remained at the level of 2.1 log CFU/g even after 4 h, representing an amount of cells corresponding to the initial inoculation. Emetic strains of *B. cereus* are mesophilic, with growth temperatures between 10 and 48 °C [41]. Compared to diarrheal strains, they show faster growth in broth at temperatures of 40–43 °C [42]. Nevertheless, cereulide formation ceases at 40 °C [41].

#### 3.3.3. Mushroom Sauce

As with the cooked rice, both strain (*p* < 0.001) and temperature (*p* = 0.033) were significant factors, including their interaction *(p* = 0.039), as only the DSM 4312 strain was able to significantly proliferate and only in samples kept at 40 °C. The growth of *B. cereus* occurred after 2.5 h of incubation at 40 °C, when the population of strain DSM 4312 exceeded 4.0 log CFU/g (Table 3). Between hours three and four of incubation, the *B. cereus* count increased by 2 log to 6.4 log CFU/g. The strain CCM 869 did not grow during the first 4 h of incubation at 40 °C; however, after 24 h its population reached 6 log CFU/g. No growth occurred at 50 and 60 °C. In contrast to the test batch of mushroom sauce prepared in the initial phase of the experiment (Section 2.1.), no isolates of naturally contaminating *B. cereus* could be obtained from the control batch after 4 h of incubation by plate counting.

#### 3.3.4. Tomato Sauce

In contrast to the previous dishes, in which the pH values exceeded 6.00, the tomato sauce had a pH value of around 4.6 after preparation (Table 2). No growth of strain DSM 4312 (Table 3), which in dishes with pH > 6.0 attained a population of over 6.0 log CFU/g after 4 h at 40 °C, was detected in this dish. Strain DSM 4312 belongs to phylogenetic group III [43], for which Webb et al. [13] stated a limit value of pH ≤ 4.3. It is possible that some constituents contained in tomatoes may also have contributed to the suppression of growth. An experiment in which the pH value of tomato sauce was increased to a value higher than 6.0 was used for clarification (Section 3.4. and Table 4).

#### 3.3.5. Ratatouille

Similarly, as in the tomato sauce, strain DSM 4312 did not grow after 4 h of incubation at 40 °C in the ratatouille dish (Table 3), although its counts were generally higher than in the case of the other reference strain (*p* = 0.010). The pH value in this dish was again around 4.6 (Table 2). Heini et al. [44] tested the growth of selected strains of *B. thuringiensis* (belonging to *B. cereus* sensu lato) in a ratatouille model. According to the authors, a pH < 6.0 inhibits the growth of *B. thuringiensis* and an increase in the population of the inoculated strains occurred only after 6 h at 22, 30, and 37 °C [44].

#### 3.3.6. Bucatini Carbonara

As with most other dishes, both strain and temperature (*p* < 0.001) were significant factors, including their interaction *(p* = 0.006), as only the DSM 4312 strain was able to significantly proliferate and only in the samples kept at 40 °C. Although the growth of both tested strains was detected in the dish bucatini carbonara (Table 3), the population of strain DSM 4312 increased by 3 log CFU/g at 40 °C within 4 h, while the increase for strain CCM 869 was only 1 log CFU/g and the increase was not statistically significant. No growth of *B. cereus* was recorded during the 4 h of the experiment at the temperatures of 50 and 60 °C.

In diseases caused by emetic strains of *B. cereus*, dishes such as spaghetti with pesto [45], spaghetti with tomato sauce [46], cooked or fried rice [47,48,49], lasagna [50], or pasta with minced meat [51] had played a role. These were mostly dishes that were consumed with a delay after preparation and the conditions of storage at cold temperatures were not observed.

Unlike cereulide, *B. cereus* enterotoxins produced in food most likely do not contribute to disease, as they are sensitive to heat, acids, and proteases [52]. Thus, only ingestion of *B. cereus* spores or viable cells leads to colonization of the small intestine, where toxins are formed between the exponential and stationary growth phases [53]. Enterotoxin production can occur in the temperature range of 10–43 °C with a temperature optimum of 32 °C; more toxin is produced at reduced oxygen levels, mimicking the conditions of the small intestine [53]. The infectious dose ranges from 10^4^ to 10^9^ CFU (vegetative cells or spores) of *B. cereus* per gram of food. The clinical symptoms of the disease are usually mild and after approximately 12–24 h the disease will disappear by itself [52].

One of the easiest ways to prevent the growth of *B. cereus* in dishes is to control the temperatures during their preparation and storage. Different methods of cooking enable the devitalization of vegetative cells and some also enable the destruction of *B. cereus* spores, depending on the temperature used and the duration of the heat treatment. One of the main causes of foodborne outbreaks caused by *B. cereus* toxins is food storage at inappropriate temperatures [39]. Hot dishes must be kept at a temperature of at least 60 °C from the time of the completion of cooking until the time of delivery. In the case of cooling after cooking, it is necessary to proceed with this process immediately after the end of the heat treatment, and the temperature should drop from 60 °C to 4 °C or less within about two hours. Cooling of the prepared dishes and its preservation at a temperature ≤4 °C is necessary from the point of view of preventing the growth of even psychrotrophic strains of *B. cereus*, although already at temperatures below 10 °C the length of the lag phase and the generation time are significantly extended [54].

The correct management of cooking and the storage of dishes are key points of HACCP plans in food service establishments [39]. Together with compliance with sanitation procedures and the effective training of personnel with regard to compliance with the principles of good production and hygienic practice, these are the basic points limiting the risk of outbreaks of foodborne diseases caused by the bacteria *B. cereus*.

### 3.4. The Growth of B. cereus in Selected Foods Following the Modification of pH

The next part of the experiment aimed to confirm the extent to which pH values influenced the growth ability of the tested strains of *B. cereus*, for which reason the pH value of mashed potatoes was reduced to 4.50 by the application of acetic acid (glacial) and, in contrast, the pH value of the tomato sauce was increased to >6.00 by the application of 10 M NaOH (Table 4). In view of the results for the growth abilities of *B. cereus* at the tested temperatures already attained, this part of the experiment was carried out only at 40 °C.

The pH modification was a significant factor (*p* < 0.001) affecting the growth of *B. cereus* in both mashed potatoes and tomato sauce. As can be seen from Table 4, the population of *B. cereus* in mashed potatoes remained suppressed during the whole incubation period in samples with modified (lowered) pH. Thus, the counts at the end of incubation for both the reference strains and the control were significantly higher in mashed potatoes with a higher pH. In contrast, an increase was recorded in the tomato sauce modified to a higher pH only for the population of the strain DSM 4312, reaching a level of 6.8 log CFU/g after 4 h. The increase detected for the strain CCM 869 was at the level of approximately 0.5 log CFU/g and not statistically significant.

According to a study by Le Marc et al. [55], the minimum pH value for the growth of *B. cereus* is dependent on the environmental temperature. The tested strains of phylogenetic group III had average pH_min_ values at 40 °C between 4.93 and 5.08 and at 20 °C between 4.66 and 4.69, whilst strains of group IV had values at 40 °C between 4.90 and 4.99 and at 20 °C between 4.70 and 4.77 [55].

To our knowledge, our study is the first to analyze the effect of pH adjustment directly in hot dishes on the growth of *B. cereus*.

## 4. Conclusions

In this study the growth of selected strains of *B. cereus* was monitored in six hot dishes during 4 h at temperatures of 40, 50, and 60 °C in order to simulate storage conditions during food delivery by courier services. The study demonstrated that a drop in food temperatures to 40 °C is risky, while no growth of *B. cereus* was detected within 4 h at 50 and 60 °C. The growth rate of *B. cereus* is conditioned not merely by environmental conditions (temperature, pH values, food composition), but also by the bacterial strains. The tested strain DSM 4312 showed good growth abilities and attained a population exceeding 6.0 log CFU/g within 4 h at 40 °C in foods with pH values > 6.0, which is a level associated with the development of foodborne diseases. It is, therefore, essential to ensure a temperature of at least 60 °C during the distribution and storage of hot dishes. Based on these findings, additional research actions will be required to analyze the growth potential of *B. cereus* associated with possible toxin formation during food storage under different conditions.

## Figures and Tables

**Figure 1 foods-14-00194-f001:**
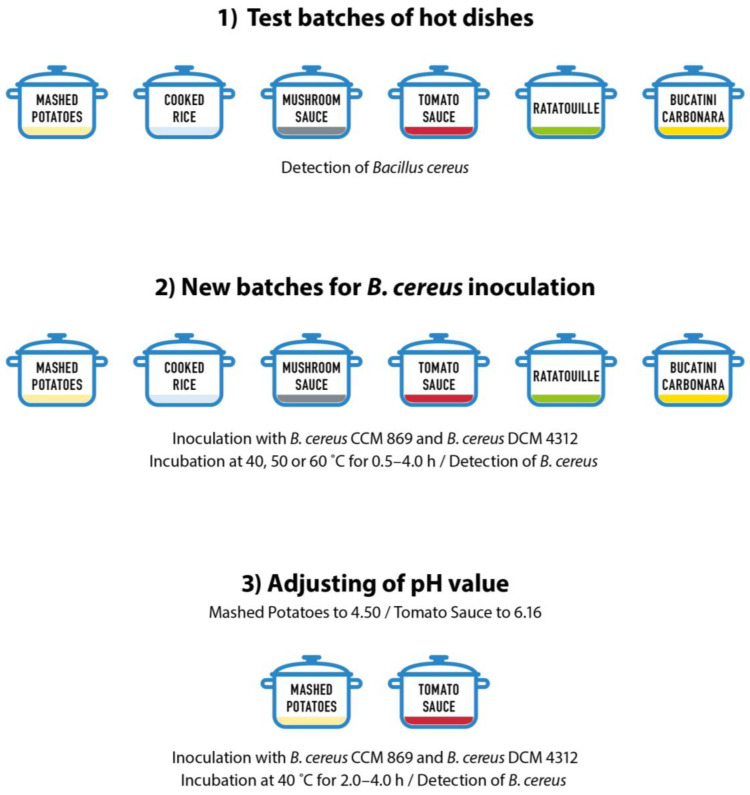
Diagram illustrating the experimental design.

**Table 1 foods-14-00194-t001:** The composition of hot ready foods.

Hot Dish	Composition
Mashed potatoes	fresh potatoes, milk
Mushroom sauce	dried mushrooms, beef broth, milk, cream, flour
Cooked rice	long-grain rice, onion
Tomato sauce	beef broth, tomato purée, carrots, celery, parsley, onion, flour
Bucatini carbonara	bucatini, egg, bacon, parmesan, olive oil, cream, onion
Ratatouille	zucchini, eggplant, peppers, tomatoes, onions, olive oil

Test batches of the given dishes were prepared for microbiological testing for the presence of *B. cereus* in the period February–March 2023.

**Table 2 foods-14-00194-t002:** Values of pH, water activity (a_w_), and proportions of salt in the hot ready meals prepared.

Hot Dish	pH	a_w_	NaCl (%)
Mashed potatoes	6.13 ± 0.00	0.973 ± 0.001	0.76 ± 0.02
Cooked rice	6.27 ± 0.02	0.973 ± 0.001	0.94 ± 0.00
Mushroom sauce	6.27 ± 0.03	0.970 ± 0.001	1.35 ± 0.28
Tomato sauce	4.63 ± 0.03	0.970 ± 0.001	0.87 ± 0.02
Ratatouille	4.63 ± 0.07	0.967 ± 0.002	1.40 ± 0.16
Bucatini carbonara	6.36 ± 0.05	0.973 ± 0.001	0.74 ± 0.02

**Table 3 foods-14-00194-t003:** The growth of *B. cereus* following the artificial contamination of hot dishes at 40, 50, and 60 °C over a period of 0–4 h (average values in log CFU/g).

		Mashed Potatoes						
t [°C]	Strain	0 h	0.5 h	1.0 h	2.0 h	2.5 h	3.0 h	4.0 h
	control	<1.7 ± 0.0 ^Aa^	<1.7 ± 0.0 ^A^	<1.7 ± 0.0 ^Aa^	1.5 ± 0.2 ^Aa^	2.5 ± 0.0 ^AB^	3.3 ± 1.0 ^ABa^	3.9 ± 0.4 ^Bbc^
40	DSM 4312	3.8 ± 0.3 ^Ab^	3.5 ± 0.0 ^A^	3.8 ± 0.3 ^Ab^	3.7 ± 0.2 ^Ab^	4.2 ± 0.0 ^AB^	4.8 ± 0.1 ^ABb^	6.4 ± 0.5 ^Bc^
	CCM 869	3.0 ± 0.1 ^ab^	3.0 ± 0.0	3.1 ± 0.1 ^ab^	3.0 ± 0.0 ^ab^	3.0 ± 0.0	3.1 ± 0.1 ^a^	4.6 ± 1.0 ^cd^
	control	<1.7 ± 0.0 ^a^	<1.7 ± 0.0	<1.7 ± 0.0 ^a^	<1.7 ± 0.0 ^a^	<1.7 ± 0.0	<1.7 ± 0.0 ^a^	<1.7 ± 0.0 ^a^
50	DSM 4312	4.0 ± 0.2 ^b^	3.7 ± 0.0	3.9 ± 0.2 ^b^	3.8 ± 0.2 ^b^	4.0 ± 0.0	4.0 ± 0.0 ^ab^	3.7 ± 0.3 ^bc^
	CCM 869	2.1 ± 0.7 ^ab^	<1.7 ± 0.0	2.2 ± 0.8 ^ab^	2.2 ± 0.8 ^ab^	2.9 ± 0.0	2.2 ± 0.8 ^a^	2.2 ± 0.8 ^ab^
	control	<1.7 ± 0.0 ^a^	<1.7 ± 0.0	<1.7 ± 0.0 ^a^	<1.7 ± 0.0 ^a^	<1.7 ± 0.0	<1.7 ± 0.0 ^a^	<1.7 ± 0.0 ^a^
60	DSM 4312	3.9 ± 0.4 ^b^	3.4 ± 0.0	3.5 ± 0.1 ^b^	3.6 ± 0.1 ^b^	4.1 ± 0.0	3.8 ± 0.0 ^ab^	3.7 ± 0.2 ^bc^
	CCM 869	2.2 ± 0.7 ^ab^	<1.7 ± 0.0	2.2 ± 0.8 ^ab^	2.2 ± 0.8 ^ab^	3.0 ± 0.0	2.3 ± 0.9 ^a^	2.2 ± 0.8 ^ab^
		**Cooked Rice**						
**t [°C]**	**Strain**	**0 h**	**0.5 h**	**1.0 h**	**2.0 h**	**2.5 h**	**3.0 h**	**4.0 h**
	control	<1.7 ± 0.0	<1.7 ± 0.0	<1.7 ± 0.0	<1.7 ± 0.0	<1.7 ± 0.0	<1.7 ± 0.0 ^a^	2.0 ± 0.6 ^a^
40	DSM 4312	2.9 ± 0.5 ^A^	3.5 ± 0.0 ^A^	3.1 ± 0.4 ^A^	3.4 ± 0.4 ^A^	3.3 ± 0.0 ^A^	4.5 ± 0.6 ^ABb^	6.0 ± 0.5 ^Bb^
	CCM 869	2.4 ± 0.1	2.5 ± 0.0	2.1 ± 0.1	1.8 ± 0.5	<1.7 ± 0.0	2.0 ± 0.6 ^a^	2.1 ± 0.7 ^a^
	control	<1.7 ± 0.0	<1.7 ± 0.0	<1.7 ± 0.0	<1.7 ± 0.0	<1.7 ± 0.0	<1.7 ± 0.0 ^a^	<1.7 ± 0.0 ^a^
50	DSM 4312	2.9 ± 0.5	3.2 ± 0.0	2.2 ± 0.2	2.7 ± 0.5	2.7 ± 0.0	2.9 ± 0.3 ^ab^	2.4 ± 1.0 ^a^
	CCM 869	1.8 ± 0.4	1.7 ± 0.0	2.8 ± 0.6	1.8 ± 0.5	1.7 ± 0.0	1.7 ± 0.0 ^a^	1.9 ± 0.5 ^a^
	control	<1.7 ± 0.0	<1.7 ± 0.0	<1.7 ± 0.0	<1.7 ± 0.0	<1.7 ± 0.0	<1.7 ± 0.0 ^a^	<1.7 ± 0.0 ^a^
60	DSM 4312	2.6 ± 0.5	3.2 ± 0.0	2.3 ± 0.8	2.8 ± 0.7	2.0 ± 0.0	2.3 ± 0.9 ^a^	2.4 ± 1.0 ^a^
	CCM 869	2.1 ± 0.1	3.2 ± 0.0	2.2 ± 2.2	1.9 ± 0.2	2.4 ± 0.0	2.1 ± 0.4 ^a^	2.1 ± 0.7 ^a^
		**Mushroom Sauce**						
**t [°C]**	**Strain**	**0 h**	**0.5 h**	**1.0 h**	**2.0 h**	**2.5 h**	**3.0 h**	**4.0 h**
	control	<1.7 ± 0.0 ^a^	<1.7 ± 0.0 ^a^	<1.7 ± 0.0 ^a^	<1.7 ± 0.0 ^a^	<1.7 ± 0.0 ^a^	<1.7 ± 0.0 ^a^	<1.7 ± 0.0 ^a^
40	DSM 4312	3.9 ± 0.2 ^Ac^	3.7 ± 0.0 ^Ab^	3.9 ± 0.2 ^Ab^	3.9 ± 0.2 ^Ab^	4.2 ± 0.0 ^Ac^	4.4 ± 0.1 ^Ac^	6.4 ± 0.0 ^Bc^
	CCM 869	2.6 ± 0.4 ^ab^	3.0 ± 0.0 ^ab^	2.8 ± 0.3 ^ab^	2.7 ± 0.4 ^ab^	2.2 ± 0.0 ^ab^	2.3 ± 0.3 ^a^	2.9 ± 0.5 ^b^
	control	<1.7 ± 0.0 ^a^	<1.7 ± 0.0 ^a^	<1.7 ± 0.0 ^a^	<1.7 ± 0.0 ^a^	<1.7 ± 0.0 ^a^	<1.7 ± 0.0 ^a^	<1.7 ± 0.0 ^a^
50	DSM 4312	3.7 ± 0.6 ^bc^	3.3 ± 0.0 ^b^	3.8 ± 0.4 ^b^	3.6 ± 0.4 ^b^	3.9 ± 0.0 ^bc^	3.6 ± 0.4 ^bc^	3.4 ± 0.6 ^b^
	CCM 869	2.5 ± 0.5 ^ab^	2.8 ± 0.0 ^ab^	2.7 ± 0.5 ^ab^	2.5 ± 0.2 ^ab^	2.2 ± 0.0 ^ab^	2.5 ± 0.5 ^ab^	2.6 ± 0.4 ^ab^
	control	<1.7 ± 0.0 ^a^	<1.7 ± 0.0 ^a^	<1.7 ± 0.0 ^a^	<1.7 ± 0.0 ^a^	<1.7 ± 0.0 ^a^	<1.7 ± 0.0 ^a^	<1.7 ± 0.0 ^a^
60	DSM 4312	3.9 ± 0.3 ^c^	3.6 ± 0.0 ^b^	3.7 ± 0.1 ^b^	3.7 ± 0.1 ^b^	3.8 ± 0.0 ^bc^	3.8 ± 0.1 ^bc^	3.7 ± 0.1 ^b^
	CCM 869	2.5 ± 0.5 ^ab^	2.8 ± 0.0 ^ab^	2.6 ± 0.4 ^ab^	2.6 ± 0.3 ^ab^	2.3 ± 0.0 ^abc^	2.6 ± 0.3 ^ab^	2.6 ± 0.4 ^ab^
		**Tomato Sauce**						
**t [°C]**	**Strain**	**0 h**	**0.5 h**	**1.0 h**	**2.0 h**	**2.5 h**	**3.0 h**	**4.0 h**
	control	<1.7 ± 0.0	<1.7 ± 0.0	<1.7 ± 0.0	<1.7 ± 0.0	<1.7 ± 0.0	<1.7 ± 0.0	<1.7 ± 0.0
40	DSM 4312	2.1 ± 0.7	2.0 ± 0.0	2.2 ± 0.8	2.5 ± 0.5	2.9 ± 0.0	2.2 ± 0.8	2.3 ± 0.6
	CCM 869	2.8 ± 0.2	3.2 ± 0.0	2.6 ± 0.5	2.3 ± 0.6	2.2 ± 0.0	2.6 ± 0.4	2.8 ± 0.3
	control	<1.7 ± 0.0	<1.7 ± 0.0	<1.7 ± 0.0	<1.7 ± 0.0	<1.7 ± 0.0	<1.7 ± 0.0	<1.7 ± 0.0
50	DSM 4312	2.4 ± 0.2	2.5 ± 0.0	2.4 ± 0.0	2.8 ± 0.2	2.5 ± 0.0	2.5 ± 0.2	1.7 ± 0.3
	CCM 869	2.1 ± 0.8	3.1 ± 0.0	2.5 ± 0.5	2.2 ± 0.7	<1.7 ± 0.0	2.2 ± 0.7	2.2 ± 0.8
	control	<1.7 ± 0.0	<1.7 ± 0.0	<1.7 ± 0.0	<1.7 ± 0.0	<1.7 ± 0.0	<1.7 ± 0.0	<1.7 ± 0.0
60	DSM 4312	2.4 ± 0.2	2.6 ± 0.0	2.2 ± 0.5	2.6 ± 0.1	1.7 ± 0.0	2.4 ± 0.4	2.0 ± 0.5
	CCM 869	2.6 ± 0.4	3.0 ± 0.0	2.3 ± 0.7	2.1 ± 0.7	1.7 ± 0.0	2.0 ± 0.6	2.0 ± 0.6
		**Ratatouille**						
**t [°C]**	**Strain**	**0 h**	**0.5 h**	**1.0 h**	**2.0 h**	**2.5 h**	**3.0 h**	**4.0 h**
	control	<1.7 ± 0.0 ^a^	<1.7 ± 0.0	<1.7 ± 0.0	<1.7 ± 0.0	<1.7 ± 0.0	<1.7 ± 0.0	<1.7 ± 0.0 ^a^
40	DSM 4312	2.8 ± 0.4 ^ab^	2.4 ± 0.0	2.8 ± 0.4	2.8 ± 0.3	3.4 ± 0.0	2.9 ± 0.3	3.0 ± 0.6 ^b^
	CCM 869	2.3 ± 0.7 ^ab^	2.8 ± 0.0	2.5 ± 0.2	2.5 ± 0.4	1.7 ± 0.0	2.2 ± 0.5	2.0 ± 0.7 ^ab^
	control	<1.7 ± 0.0 ^a^	<1.7 ± 0.0	<1.7 ± 0.0	<1.7 ± 0.0	<1.7 ± 0.0	<1.7 ± 0.0	<1.7 ± 0.0 ^a^
50	DSM 4312	3.0 ± 0.4 ^b^	2.3 ± 0.0	2.6 ± 0.3	2.8 ± 0.4	3.1 ± 0.0	2.9 ± 0.2	2.6 ± 0.5 ^ab^
	CCM 869	2.7 ± 0.1 ^ab^	2.8 ± 0.0	2.2 ± 0.5	2.6 ± 0.4	2.0 ± 0.0	2.0 ± 0.6	2.2 ± 0.2 ^ab^
	control	<1.7 ± 0.0 ^a^	<1.7 ± 0.0	<1.7 ± 0.0	<1.7 ± 0.0	<1.7 ± 0.0	<1.7 ± 0.0	<1.7 ± 0.0 ^a^
60	DSM 4312	2.4 ± 0.0 ^ab^	<1.7 ± 0.0	2.0 ± 0.3	2.1 ± 0.4	2.6 ± 0.0	1.9 ± 0.2	2.2 ± 0.1 ^ab^
	CCM 869	2.1 ± 0.7 ^ab^	2.8 ± 0.0	2.2 ± 0.2	2.0 ± 0.3	<1.7 ± 0.0	2.0 ± 0.6	1.8 ± 0.5 ^ab^
		**Bucatini Carbonara**						
**t [° C]**	**Strain**	**0 h**	**0.5 h**	**1.0 h**	**2.0 h**	**2.5 h**	**3.0 h**	**4.0 h**
	control	<1.7 ± 0.0 ^a^	<1.7 ± 0.0	<1.7 ± 0.0 ^a^	<1.7 ± 0.0 ^a^	<1.7 ± 0.0 ^a^	<1.7 ± 0.0 ^a^	<1.7 ± 0.0 ^a^
40	DSM 4312	3.6 ± 0.1 ^Ab^	3.7 ± 0.0 ^AB^	3.7 ± 0.0 ^ABb^	4.4 ± 0.3 ^ABb^	5.3 ± 0.0 ^ABb^	5.4 ± 0.2 ^BCc^	6.7 ± 0.2 ^Cc^
	CCM 869	2.8 ± 0.4 ^ab^	3.2 ± 0.0	2.4 ± 0.7 ^ab^	2.7 ± 0.5 ^ab^	2.2 ± 0.0 ^a^	3.0 ± 0.5 ^ab^	4.1 ± 0.7 ^b^
	control	<1.7 ± 0.0 ^a^	<1.7 ± 0.0	<1.7 ± 0.0 ^a^	<1.7 ± 0.0 ^a^	<1.7 ± 0.0 ^a^	<1.7 ± 0.0 ^a^	<1.7 ± 0.0 ^a^
50	DSM 4312	3.5 ± 0.1 ^b^	3.5 ± 0.0	3.5 ± 0.1 ^b^	3.8 ± 0.4 ^b^	4.5 ± 0.0 ^b^	4.1 ± 0.7 ^bc^	3.8 ± 0.7 ^b^
	CCM 869	2.3 ± 0.9 ^ab^	3.0 ± 0.0	2.2 ± 0.8 ^ab^	2.3 ± 0.9 ^ab^	<1.7 ± 0.0 ^a^	2.5 ± 0.5 ^ab^	2.3 ± 0.9 ^a^
	control	<1.7 ± 0.0 ^a^	<1.7 ± 0.0	<1.7 ± 0.0 ^a^	<1.7 ± 0.0 ^a^	<1.7 ± 0.0 ^a^	<1.7 ± 0.0 ^a^	<1.7 ± 0.0 ^a^
60	DSM 4312	3.5 ± 0.1 ^b^	3.3 ± 0.0	3.2 ± 0.2 ^ab^	3.1 ± 0.1 ^ab^	3.3 ± 0.0 ^ab^	3.1 ± 0.1 ^ab^	3.0 ± 0.3 ^ab^
	CCM 869	2.2 ± 0.8 ^ab^	3.2 ± 0.0	2.3 ± 0.6 ^ab^	2.2 ± 0.8 ^ab^	<1.7 ± 0.0 ^a^	2.3 ± 0.3 ^ab^	2.3 ± 0.6 ^a^

Letters ^a–d^ mark statistically significant differences (*p* < 0.05) within the column (for each dish separately). Letters ^A–C^ mark statistically significant differences (*p* < 0.05) within the row.

**Table 4 foods-14-00194-t004:** The growth of *B. cereus* following the artificial contamination of mashed potatoes and tomato sauce at 40 °C following the modification of pH values (average values in log CFU/g).

		Mashed Potatoes						
		pH	a_w_	NaCl (%)	0 h	2.0 h	3.0 h	4.0 h
control	original	6.13 ± 0.00	0.973 ± 0.001	0.76 ± 0.02	<1.7 ^Aa^	1.5 ± 0.2 ^Aab^	3.3 ± 1.0 ^Bbc^	3.9 ± 0.4 ^Bbc^
	mod.	4.50 ± 0.02	0.971 ± 0.003	1.05 ± 0.11	<1.7 ^a^	<1.7 ^a^	<1.7 ^a^	<1.7 ^a^
DSM 4312	original	6.13 ± 0.00	0.973 ± 0.001	0.76 ± 0.02	3.8 ± 0.3 ^Ac^	3.7 ± 0.2 ^Ac^	4.8 ± 0.1 ^ABc^	6.4 ± 0.5 ^Bd^
	mod.	4.50 ± 0.02	0.971 ± 0.003	1.05 ± 0.11	2.6 ± 0.2 ^abc^	2.6 ± 0.2 ^abc^	2.7 ± 0.1 ^ab^	3.0 ± 0.1 ^ab^
CCM 869	original	6.13 ± 0.00	0.973 ± 0.001	0.76 ± 0.02	3.0 ± 0.1 ^Abc^	3.0 ± 0.0 ^Abc^	3.1 ± 0.1 ^Ab^	4.6 ± 1.0 ^Bc^
	mod.	4.50 ± 0.02	0.971 ± 0.003	1.05 ± 0.11	1.8 ± 0.4 ^ab^	1.7 ± 0.3 ^ab^	2.2 ± 0.1 ^ab^	1.7 ± 0.3 ^a^
		**Tomato Sauce**						
		**pH**	**a_w_**	**NaCl (%)**	**0 h**	**2.0 h**	**3.0 h**	**4.0 h**
control	mod.	6.16 ± 0.05	0.966 ± 0.005	1.00 ± 0.11	<1.7 ^a^	<1.7 ^a^	<1.7 ^a^	<1.7 ^a^
	original	4.63 ± 0.03	0.970 ± 0.001	0.87 ± 0.02	<1.7 ^a^	<1.7 ^a^	<1.7 ^a^	<1.7 ^a^
DSM 4312	mod.	6.16 ± 0.05	0.966 ± 0.005	1.00 ± 0.11	4.1 ± 0.1 ^Ab^	4.8 ± 0.1 ^ABb^	5.7 ± 0.2 ^ABb^	6.8 ± 0.1 ^Bb^
	original	4.63 ± 0.03	0.970 ± 0.001	0.87 ± 0.02	2.1 ± 0.7 ^a^	2.5 ± 0.5 ^a^	2.2 ± 0.8 ^a^	2.3 ± 0.6 ^a^
CCM 869	mod.	6.16 ± 0.05	0.966 ± 0.005	1.00 ± 0.11	2.2 ± 0.2 ^a^	1.7 ± 0.3 ^a^	2.2 ± 0.5 ^a^	2.9 ± 0.5 ^a^
	original	4.63 ± 0.03	0.970 ± 0.001	0.87 ± 0.02	2.8 ± 0.2 ^ab^	2.3 ± 0.6 ^a^	2.6 ± 0.4 ^a^	2.8 ± 0.3 ^a^

Letters ^a–c^ mark statistically significant differences within the column (*p* < 0.05) (for each dish separately). Letters ^A–B^ mark statistically significant differences (*p* < 0.05) within the row. mod.—modification.

## Data Availability

The original contributions presented in this study are included in the article and Appendix A. Further inquiries can be directed to the corresponding author.

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
