# Peer review of "The Germination and Growth of Two Strains of Bacillus cereus in Selected Hot Dishes After Cooking"

_foods, 2025, doi:10.3390/foods14020194_

Round 1
Reviewer 1 Report (Previous Reviewer 5)
Comments and Suggestions for Authors
Dear authors,
I read your work carefully and it seemed to me to be much improved from the previous description. The experiment description is accurate and the evaluation of the results congruous.
Best regards.
Author Response
Dear Reviewer 1,
Very thank you for your review report.
Kind regards,
Marta Dušková, Ph.D.
Corresponding Author
University of Veterinary Sciences Brno
Faculty of Veterinary Hygiene and Ecology
Department of Animal Origin Food & Gastronomic Sciences
Address: Palackého tř. 1946/1, 612 42 Brno, Czech Republic
E-mail: duskovam@vfu.cz

Reviewer 2 Report (Previous Reviewer 2)
Comments and Suggestions for Authors
The manuscript has been improved and now can be accepted for publication in its present form.
Author Response
Dear Reviewer 2,
Very thank you for your review report.
Kind regards,
Marta Dušková, Ph.D.
Corresponding Author
University of Veterinary Sciences Brno
Faculty of Veterinary Hygiene and Ecology
Department of Animal Origin Food & Gastronomic Sciences
Address: Palackého tř. 1946/1, 612 42 Brno, Czech Republic
E-mail: duskovam@vfu.cz

This manuscript is a resubmission of an earlier submission. The following is a list of the peer review reports and author responses from that submission.
Round 1
Reviewer 1 Report
Comments and Suggestions for Authors
The aim of this study was to assess the germination and growth of two B. cereus strains following the artificial inoculation with spores of six selected hot dished stored at different temperatures for different time. There are some comments.
General comments:
1. Two B. cereus strains CCM869 AND DSM4312 were used, what are the phylogenetic groups of the strains?
2. Suspected colonies of B. cereus were identified by MALDI-TOF MS and PCR? What is the difference in the purpose of using the two methods? Both were used to identify B. cereus? For PCR detection, gyrB was amplified, the author determined the isolates without sequencing of gyrB?
3. The description of the results section was confusing, such as 3.2 and 3.3.
4. The quality of figures and tables was poor and please improve their quality.
Specific comments:
1. line 77: an indefinite article should added.
2. line 99: Five hundred microliters
3. line 147: After the matrix dried
4. line 183: all data were just entered into spreadsheets without further differential analysis?
5. line 188: “As shown in Table 2, all the …”
6. line 189: what is the limit salt for the growth of B. cereus?
7. line 192: B. cereus belongs to phylogenetic group VI? Or B. cereus were divided into several groups?
8. line 207: What is the basis for the detection limit?
Comments on the Quality of English LanguageEnglish usage should be improved.
Author Response
Responses to reviewer 1
We thank the reviewer 1 for his valuable comments and advice, which helped improve the quality of our manuscript.
- Two B. cereus strains CCM869 AND DSM4312 were used, what are the phylogenetic groups of the strains?
Information about the phylogenetic group of strain DSM 4312 is in the text (lines 317-318), the classification of strain CCM 869 cannot be found.
- Suspected colonies of B. cereus were identified by MALDI-TOF MS and PCR? What is the difference in the purpose of using the two methods? Both were used to identify B. cereus? For PCR detection, gyrB was amplified, the author determined the isolates without sequencing of gyrB?
Mass spectrometry measurements were performed using an UltrafleXtreme instrument (Bruker Daltonik, Germany) operated in the linear positive-ion mode using FlexControl 3.4 software. Mass spectra were processed using BioTyper software (version 3.0; Bruker Daltonik). The identification results were expressed by BioTyper log(scores) indicating the similarity of the unknown MALDI-TOF MS profile to BioTyper database entries (version 10.0; 9,607 entries). A BioTyper log (score) greater than 2.0 indicates a secure genus identification and probable or highly probable species identification. In order to confirm the species identification, a PCR method was performed. The PCR method was aimed at detecting the gyrB gene. Further sequencing was not necessary and was therefore not performed.
- The description of the results section was confusing, such as 3.2 and 3.3.
We have made adjustments in chapters 3.2. and 3.3. In addition, we have supplemented the manuscript with a diagram illustrating the experimental design for better orientation in the text.
- The quality of figures and tables was poor and please improve their quality.
We decided to replace figures 1-6 with a table expressing all CFU/g values. Although the figures allow a quick orientation in the development of the B. cereus population during the incubation of the hot dishes, the tabular representation has a higher informative value and therefore we decided to replace the figures with a table.
Specific comments:
- line 77: an indefinite article should added.
The text has been corrected.
- line 99: Five hundred microliters
The phrase Five hundred microliters was added to the beginning of the sentence.
- line 147: After the matrix dried
The correction was made in the manuscript text.
- line 183: all data were just entered into spreadsheets without further differential analysis?
Suspect colonies on the plates were counted, confirmed and identified. The colony forming units per gram (CFU/g) value was calculated using the formula (weighted average of two consecutive dilutions):
|
N = |
S c |
|
V (n1 + 0.1n2) d |
S c the sum of characteristic colonies counted after identification
V the amount of inoculum added into Petri dish (1 mL or 0.2 mL)
n1 the number of Petri dishes retained in the 1st dilution
n2 the number of Petri dishes retained in the 2nd dilution
d the dilution factor corresponding to the 1st dilution
- line 188: “As shown in Table 2, all the …”
The correction was made in the manuscript text.
- line 189: what is the limit salt for the growth of B. cereus?
It is described on lines 198-200.
- line 192: B. cereus belongs to phylogenetic group VI? Or B. cereus were divided into several groups?
The text has been added and explains the question asked.
- line 207: What is the basis for the detection limit?
The detection limit is based on the amount of inoculum spread on the agar in the Petri dish and the dilution used.

Reviewer 2 Report
Comments and Suggestions for Authors
This manuscript assessed the germination and growth of two strains of Bacillus cereus following the artificial inoculation with spores in hot dishes stored at different temperatures. The reasearch tpoic is interesting, however the experiments design seemes not rigorous enough. why authors ajusted the pH of two dishes,not all six dishes?
Also, sufficient statistical analysis of experiment data was not performed.
Additionally, the presentation of the results is written well. For instance, several sections do not present and analyse the results of this study, instead appearing to be a comprehensive literature review and discussion, eg.section 3.3.
All the figures are of a poor quality and do not meet the standards required for publication.
Therefore, taken together, the novelty and writing of this manuscript do not yet meet the requirements for journal publication.
Author Response
Responses to reviewer 2
We thank the reviewer 2 for his valuable comments and advice, which helped improve the quality of our manuscript.
This manuscript assessed the germination and growth of two strains of Bacillus cereus following the artificial inoculation with spores in hot dishes stored at different temperatures. The reasearch tpoic is interesting, however the experiments design seemes not rigorous enough. why authors ajusted the pH of two dishes,not all six dishes?
Of the 6 hot dishes prepared, 4 had pH values > 6.0 and 2 had pH values around 4.0. In these 2 dishes, no growth of B. cereus occurred at 40 °C, while in the other 4 with pH values > 6.0, growth occurred. Therefore, we first adjusted the pH to > 6.0 in the tomato sauce, because due to the structure of the dish, it was easier than with ratatouille. When we demonstrated the growth of B. cereus in tomato sauce with adjusted pH > 6.0 in 2 experiments, it can be assumed based on predictive microbiology that this would also be the case in ratatouille. Conversely, we wanted to demonstrate that even in foods with a standard pH value > 6.0, B. cereus would not grow after adjusting the pH to values around 4.0. We chose mashed potatoes because it was possible to adjust the pH value more precisely (due to the structure) than, for example, in cooked rice or bucatini carbonara.
Also, sufficient statistical analysis of experiment data was not performed.
We do not believe that complex statistical analysis are necessary in evaluating experiments monitoring the growth of indicator bacteria. Our analyses confirmed under which conditions B. cereus can grow in hot dishes (temperature 40 °C, pH > 6.0, within a few hours) and under which conditions it does not grow (temperatures 50 and 60 °C, pH around 4.0) and that growth also depends on the specific strain of B. cereus.
Additionally, the presentation of the results is written well. For instance, several sections do not present and analyse the results of this study, instead appearing to be a comprehensive literature review and discussion, eg.section 3.3.
The correction was made in the manuscript text.
All the figures are of a poor quality and do not meet the standards required for publication.
We decided to replace figures 1-6 with a table expressing all CFU/g values. Although the figures allow a quick orientation in the development of the B. cereus population during the incubation of the hot dishes, the tabular representation has a higher informative value and therefore we decided to replace the figures with a table.
Therefore, taken together, the novelty and writing of this manuscript do not yet meet the requirements for journal publication.
This study describes the results of a project to monitor the risk of a delivery of hot dishes (from the perspective of sporogenic bacteria) by distribution services to end customers. The growth of B. cereus in selected dishes was monitored at specific temperatures for a certain period of time. We do not claim that similar experiments have not been carried out in the past. But certainly not simultaneously in 6 hot dishes. Moreover, the issue of B. cereus as a cause of foodborne diseases is becoming increasingly serious. In the EU, it is among the most common causes of foodborne outbreaks. The results of our study highlight that: i) B. cereus can grow in hot dishes as a natural microbiota that has survived cooking during food preparation; ii) B. cereus cannot grow in hot dishes at temperatures of 50°C or more, which is crucial for preventive measures; iii) the growth of B. cereus under the experimental conditions used is dependent on the strain present.

Reviewer 3 Report
Comments and Suggestions for Authors
The paper "The growth of Bacillus cereus after spore inoculation of selected hot dishes depending on temperature and time" is relatively comprehensive and sound. However, there is a lack of novelty since the problem is already well known, see e.g. ref 11 and several other publications from USFDA.
Fig 1C: Clearly the staring point for the 40 degrees C is approx 1 log higher, this may have biased the results. A major limitation is the lack of replicates.
Table 3: In mashed potatoes the CCm 869 count is 0,5 log higher at 3 h vs at 4 h, and similarly in tomato sauce where it goes down at 2 h and up again at 3 h. This is not logical and there is no evidence of reproducibility. Again, a major limitation is the lack of replicates.
To improve the impact of this paper, a comprehensive and general risk analysis may be included in the discussion.
Moreover, there is a lack of information regarding the strains used in this study, and there is no information about the purity of the spore suspensions used. Replicates with new spore crops need to be included.
Minor comments:
L123: For how long time were the samples homogenized?
Author Response
Responses to reviewer 3
We thank the reviewer 3 for his valuable comments and advice, which helped improve the quality of our manuscript.
The paper "The growth of Bacillus cereus after spore inoculation of selected hot dishes depending on temperature and time" is relatively comprehensive and sound. However, there is a lack of novelty since the problem is already well known, see e.g. ref 11 and several other publications from USFDA.
Fig 1C: Clearly the staring point for the 40 degrees C is approx 1 log higher, this may have biased the results. A major limitation is the lack of replicates.
We decided to replace figures 1-6 with a table expressing all CFU/g values. Although the figures allow a quick orientation in the development of the B. cereus population during the incubation of the hot dishes, the tabular representation has a higher informative value and therefore we decided to replace the figures with a table. The experiment was performed in duplicate with an interval of several weeks. Given the complexity of the experiment in terms of preparation and analyses, we did not perform multiple repetitions. Given that repeating the experiment confirmed the results of the first series, we did not consider it necessary.
Table 3: In mashed potatoes the CCM 869 count is 0,5 log higher at 3 h vs at 4 h, and similarly in tomato sauce where it goes down at 2 h and up again at 3 h. This is not logical and there is no evidence of reproducibility. Again, a major limitation is the lack of replicates.
Mashed potatoes with a pH of 4.50 did not allow B. cereus to grow. The values shown in Table 4 (Table 3 in previous version of manuscript) only indicate spore survival, not growth. And differences of the order of 0.5 log are within the limit of detection.
To improve the impact of this paper, a comprehensive and general risk analysis may be included in the discussion.
Thank you for your comment. The risk analysis from the perspective of B. cereus is based on two variables: 1) the temperature of the food is crucial. In the case of hot dishes (according to the results of this study), the minimum temperature (that blocks the growth of B. cereus) is 50 °C, the legislation of the nation states can regulate the conditions. In the Czech Republic, at least 60 °C, for example in Spain 63 °C. 2) It is also a question of the pH value.
But from the perspective of food service establishments, no one monitors the pH value of the hot dishes. Therefore, it is essential to adhere to the correct temperature regime.
Moreover, there is a lack of information regarding the strains used in this study, and there is no information about the purity of the spore suspensions used. Replicates with new spore crops need to be included.
Thank you for the reminder. Available information on the collection strains used is listed in the manuscript and some has been added. Only pure spore suspensions were used. The control of spore suspensions was carried out microscopically and by culture. Due to the complexity of the experiment, analyses were only performed twice.
Minor comments:
L123: For how long time were the samples homogenized?
The text has been added.

Reviewer 4 Report
Comments and Suggestions for Authors
Line 19-20: I suggest you to modify or eliminate from keywords, the foods name. You should use relevant general words, such as ready-to-eat meals or hot ready meals.
Line 95-117: Please provide a bibliographic source or standard regulation for the preparation of the initial suspension
Line 138: Please correct the incubation temperature, there are mentioned two, but only the 44 degrees Celsius is correct.
Line 153-154: Please modify the text format
In the results part, where you obtain a bacteria growth in the control batches, I suggest you to explain in a few sentences, why your results are relevant even with an initial contamination. Also, the figure 3-6 are not mentioned in text.
Line 353-354: The sentence is incomplete. Please rewrite it.54
Author Response
Responses to reviewer 4
We thank the reviewer 4 for his valuable comments and advice, which helped improve the quality of our manuscript.
Line 19-20: I suggest you to modify or eliminate from keywords, the foods name. You should use relevant general words, such as ready-to-eat meals or hot ready meals.
The text has been corrected.
Line 95-117: Please provide a bibliographic source or standard regulation for the preparation of the initial suspension
The bibliographic source has been added.
Line 138: Please correct the incubation temperature, there are mentioned two, but only the 44 degrees Celsius is correct.
The text has been corrected.
Line 153-154: Please modify the text format
The text format has been corrected.
In the results part, where you obtain a bacteria growth in the control batches, I suggest you to explain in a few sentences, why your results are relevant even with an initial contamination. Also, the figure 3-6 are not mentioned in text.
Thank you for your comment. When simulating the conditions of spore germination in real foods, natural contamination cannot be ruled out (especially when spores germinate only after several hours). We believe that the results are relevant despite this fact.
We decided to replace figures 1-6 with a table expressing all CFU/g values. Although the figures allow a quick orientation in the development of the B. cereus population during the incubation of the hot dishes, the tabular representation has a higher informative value and therefore we decided to replace the figures with a table.
Line 353-354: The sentence is incomplete. Please rewrite it.54
The text has been corrected.

Reviewer 5 Report
Comments and Suggestions for Authors
Dear authors,
Your paper has a good scientific sounds, is well described and the results are clear. However, you need to check better the correspondence between figures and text, as some figures are not correctly cited in the text.
Author Response
Responses to reviewer 5
We thank the reviewer 5 for his valuable comments and advice, which helped improve the quality of our manuscript.
Your paper has a good scientific sounds, is well described and the results are clear. However, you need to check better the correspondence between figures and text, as some figures are not correctly cited in the text.
The correction was made in the manuscript text.

Round 2
Reviewer 1 Report
Comments and Suggestions for Authors
The authors have revised the manuscript carefully.
Author Response
Responses to Reviewer 1
Dear Reviewer 1,
Very thank you for your review report.
Kind regards,
Marta Dušková, Ph.D.
Corresponding Author
University of Veterinary Sciences Brno
Faculty of Veterinary Hygiene and Ecology
Department of Animal Origin Food & Gastronomic Sciences
Address: Palackého tř. 1946/1, 612 42 Brno, Czech Republic
E-mail: duskovam@vfu.cz

Reviewer 2 Report
Comments and Suggestions for Authors
The authors have addressed the vast majority of the reviewers' concerns. However, I strongly recommend that the most basic statistical analyses, including ANOVA and multiple comparisons of means, be performed. Without these revision, I cannot accept the manuscript for publication and dissemination to the scientific community.
Author Response
Responses to Reviewer 2
We thank the Reviewer 2 for his valuable comments and advice, which helped improve the quality of our manuscript.
“The authors have addressed the vast majority of the reviewers' concerns. However, I strongly recommend that the most basic statistical analyses, including ANOVA and multiple comparisons of means, be performed. Without these revision, I cannot accept the manuscript for publication and dissemination to the scientific community.”
Statistical analysis has been added.
Kind regards,
Marta Dušková, Ph.D.
Corresponding Author
University of Veterinary Sciences Brno
Faculty of Veterinary Hygiene and Ecology
Department of Animal Origin Food & Gastronomic Sciences
Address: Palackého tř. 1946/1, 612 42 Brno, Czech Republic
E-mail: duskovam@vfu.cz

Reviewer 3 Report
Comments and Suggestions for Authors
Lack of novelty
Author Response
Responses to Reviewer 3
Dear Reviewer 3,
Thank you for your review report.
Kind regards,
Marta Dušková, Ph.D.
Corresponding Author
University of Veterinary Sciences Brno
Faculty of Veterinary Hygiene and Ecology
Department of Animal Origin Food & Gastronomic Sciences
Address: Palackého tř. 1946/1, 612 42 Brno, Czech Republic
E-mail: duskovam@vfu.cz
